# Antioxidants Supplementation Reduces Ceramide Synthesis Improving the Cardiac Insulin Transduction Pathway in a Rodent Model of Obesity

**DOI:** 10.3390/nu13103413

**Published:** 2021-09-28

**Authors:** Katarzyna Hodun, Klaudia Sztolsztener, Adrian Chabowski

**Affiliations:** Department of Physiology, Medical University of Bialystok, 15-089 Bialystok, Poland; klaudia.sztolsztener@umb.edu.pl (K.S.); adrian@umb.edu.pl (A.C.)

**Keywords:** n-acetylcysteine, α-lipoic acid, sphingolipid, ceramide, insulin resistance, left ventricle, cardiovascular diseases

## Abstract

Obesity-related disruption in lipid metabolism contributes to cardiovascular dysfunction. Despite numerous studies on lipid metabolism in the left ventricle, there is no data describing the influence of n-acetylcysteine (NAC) and α-lipoic acid (ALA), as glutathione precursors, on sphingolipid metabolism, and insulin resistance (IR) occurrence. The aim of our experiment was to evaluate the influence of chronic antioxidants administration on myocardial sphingolipid state and intracellular insulin signaling as a potential therapeutic strategy for obesity-related cardiovascular IR. The experiment was conducted on male Wistar rats fed a standard rodent chow or a high-fat diet with intragastric administration of NAC or ALA for eight weeks. Cardiac and plasma sphingolipid species were assessed by high-performance liquid chromatography (HPLC). The proteins expressed from sphingolipid and insulin signaling pathways were determined by Western blot. Antioxidant supplementation markedly reduced ceramide accumulation by lowering the expression of selected proteins from the sphingolipid pathway and simultaneously increased the myocardial sphingosine-1-phosphate level. Moreover, NAC and ALA augmented the expression of GLUT4 and the phosphorylation state of Akt (Ser473) and GSK3β (Ser9), which improved the intracellular insulin transduction pathway. Based on our results, we may postulate that NAC and ALA have a beneficial influence on the cardiac ceramidose under IR conditions.

## 1. Introduction

In recent years, an increase in the number of diseases associated with obesity has been estimated to be a tremendous problem in modern societies. These alarming numbers are a consequence of overnutrition, especially high-fat consumption and sedentary lifestyle, which finally leads to increased incidence of obesity as an important factor in the development of various metabolic and cardiovascular health consequences [1].

It is well known that increased lipid availability causes changes in cardiac myocytes’ lipid metabolism (β-oxidation and esterification) which results in its serious dysfunction [2]. Enhanced fatty acids (FAs) delivery provokes excessive triacylglycerol (TAG) synthesis and storage of FAs in peripheral tissues such as the heart [3]. This increased cardiac lipid storage can be a lipid overload indicator and a marker of impaired FAs metabolism [4,5]. Although the rise of intracellular lipid accumulation is not only the conversion of FAs into TAG but also esterifying them into more bioactive lipid pools such as sphingolipids and their essential fractions, especially sphinganine (SFA), ceramide (CER), and sphingosine (SFO)) [5]. It is known that the increased content of ventricular major sphingolipid—ceramide enhances lipotoxicity and as an apoptotic mediator promotes cell death [3,6]. Many studies reported that ceramide deposition also alters insulin signaling and insulin-dependent metabolism of lipid and glucose through the activation of protein phosphatase 2 (PP2A) that ultimately results in protein kinase B (Akt/PKB) dephosphorylation. This increase in dephosphorylation of the Akt pathway has an inhibitory impact on glucose uptake via reducing glucose transporter 4 (GLUT 4) recruitment to the membrane [5,6,7,8,9,10]. Moreover, the accumulation of CER contributes to the impairment of mitochondrial oxidative functioning and also exacerbates lipid deposition and IR [11]. By these mechanisms, systemic insulin resistance is related to lipotoxicity in metabolic tissues [11,12]. The best solution to counteract lipid disturbances seems to be a diminishment in FAs availability and decreased lipid accumulation in various tissues which have a beneficial effect on the aforementioned obesity-mediated disorders [13,14]. It has been described that the administration of antioxidants, i.e., α-lipoic acid (ALA) and n-acetylcysteine (NAC), effectively influences lipid metabolism and shows an anti-obesity effect. Treatment with ALA exerted a notable reduction of mice’ body mass and content of visceral fat in a lipid overload condition [13]. Moreover, α-lipoic acid and n-acetylcysteine enhanced β-oxidation of fatty acids and decreased intracellular TAG accumulation in muscle cells and adipose tissue, respectively [13,15]. Indirectly, through mentioned changes in the FAs oxidation and then TAG content, ALA and NAC may alter the sphingolipid pathway leading to attenuate especially the deposition of ceramide induced by lipid overload conditions. Hence, alteration in sphingolipid metabolism (e.g., ceramide) may be a therapeutic strategy for obesity-related cardiovascular IR. Thus, the goal of this study was to elucidate changes in cardiac sphingolipids, especially ceramide synthesis and insulin resistance development after antioxidants (NAC and ALA) treatment in order to improve heart insulin resistance.

## 2. Materials and Methods

### 2.1. Animals and Experimental Protocol

The experiment was carried out on male Wistar rats with a body weight of approximately 50–70 g and 4 weeks of age at the beginning of the experiment. The experiment was conducted in two sets, i.e., the first set—NAC and the second—ALA, each at a different time point (40 rats per every experimental set; 80 rats for all groups in the experiment). Throughout the entire duration of the study, the rats were kept in standard laboratory animal living conditions: plastic autoclavable cages, 22 ± 2 °C air temperature, 12 h light/dark rhythm, and illimitable access to water and chow. After one week of acclimatization to the new environment, rats were randomly divided in equal numbers of 10 individuals to each of four groups in the NAC experiment and 10 individuals to each of four groups in the ALA experiment: Control group—rats fed a standard rodent chow consisting of 65.5% carbohydrates, 24.2% proteins and 10.3% fats (Agropol, Motycz, Poland); HFD group—rats receiving a high-fat diet (HFD) consisting of 59.8% fat, 20.1% proteins and 20.1% carbohydrates (Research Diet, D12492, USA); Control + NAC group/Control + ALA group—rats fed the above-described standard diet plus n-acetylcysteine or α-lipoic acid; HFD + NAC group/HFD + ALA group—rats fed the high-fat diet as well as n-acetylcysteine or α-lipoic acid. The experiment was conducted for eight weeks. Once-daily, each morning between 8–9 am, NAC (at a dose of 500 mg/kg body weight, Sigma-Aldrich, A9165) and ALA (at a dose of 30 mg/kg body weight, Sigma-Aldrich, PHR2561) were dissolved in saline solution and immediately applied intragastrically by gastric gavage to rats from appropriate groups (Figure 1). The rats from Control and HFD groups received only saline solution. The intragastric administration of antioxidants ensured that rats obtained full dose calculated for body weight. The amount of antioxidants administered was adjusted according to the current body weight of rats and it was controlled every two days. ALA and NAC concentrations were established based on available data at doses that provide satisfactory antioxidant effect and eliminates the risk of toxicity in Wistar rats [16,17,18].

At the end of the experiment, fasted overnight, the rats were anesthetized by intraperitoneal phenobarbital injection (80 mg/kg of body weight). The left ventricle was excised and immediately frozen in liquid nitrogen using precooled clamped aluminum tongs. Blood was collected in tubes containing EDTA and centrifuged in order to separate plasma. The collected samples of the left ventricle and plasma were stored at −80 °C until further analysis. The investigation was approved by the Ethical Committee for Animal Testing in Bialystok (No. 21/2017).

### 2.2. Plasma Measurements

The plasma content of glucose and insulin was determined by the use of the colorimetric reagent kit (Glucose Colorimetric Assay Kit II; BioVision Inc., Milpitas, CA, USA) and ELISA kit (Rat Insulin ELISA Kit; Mercodia, Uppsala, Sweden), respectively, according to the manufacturer’s protocols. The absorbance was measured spectrophotometrically at 450 nm using a microplate reader (Synergy H1TM, BioTek Instruments, Winooski, VT, USA). Then, the values of parameters were calculated from the obtained standard curves. Moreover, HOMA-IR (homeostasis model of assessment for insulin resistance) was calculated based on the glucose and insulin values (HOMA-IR = (FPG × FPI/22.5)).

### 2.3. Analysis of Sphingoid Bases

The plasma and cardiac content of sphingosine (SFO), sphinganine (SFA), sphingosine-1-phosphate (S1P), and sphinganine-1-phosphate (SA1P) were measured using high-performance liquid chromatography method (HPLC), as previously described in detail by Min et al. [19]. Firstly, the left ventricle samples were homogenized in a buffer containing 1 M NaCl and 25 mM HCl. Next, into preadded internal standards (10 pmol of C17-sphingosine as well as 30 pmol of C17-sphingosine-1-phosphate, Avanti Polar Lipids, Alabaster, AL, USA) and acidified methanol the samples, i.e., tissue and plasma were ultrasonicated in ice for one minute. The extraction of lipids was performed using chloroform (CHCl_3_), 1 M NaCl, and 3 N NaOH. After that, the alkaline aqueous phase with phosphorylated sphingoid bases was pipetted out into a fresh tube. The residual of sphingoid base-1-phosphates in the chloroform phase was extracted again two times with methanol/1 M NaCl (1:1, *v*/*v*) mixture and 3 N NaOH. All the aqueous fractions were mixed and then alkaline phosphatase was added. In addition, to improve the extraction process CHCl_3_ was carefully inserted into the bottom of the reaction mixture. In this procedure, the content of sphingosine-1-phosphate and sphinganine-1-phosphate were measured indirectly after conversion into sphingosine and sphinganine, respectively, by alkaline phosphatase (Sigma-Aldrich). The chloroform fractions containing dephosphorylated sphingoid bases and free sphingosine and sphinganine were washed using alkaline water. The CHCl_3_ layer was transferred into a fresh tube and dried under a nitrogen stream. The residue of dried lipid was dissolved in ethanol and then the lipid solution was mixed with an o-phthalaldehyde reagent (OPA). The OPA-derivates were analyzed by HPLC (ProStar, Varian Inc. Palo Alto, CA, USA) equipped with a fluorescence detector as well as a C18 reversed-phase column (OmniSpher 5, Varian Inc. 4.6 × 150 mm). The isocratic eluent contained acetonitrile (Merck, Darmstadt, Germany) and water (9:1, *v*/*v*). The temperature of the column was maintained at 30 °C.

### 2.4. Analysis of Ceramide

The content of ceramide was determined by the procedure previously described by Baranowski et al. [20]. A small aliquot of the above-mentioned chloroform phase was pipetted out into a fresh tube containing internal standard [40 pmol of N-palmitoyl-D-erythro-sphingosine (C17 base)]. Samples were evaporated with a nitrogen stream and then the dried lipid was dissolved in 1 M KOH in 90% methanol. After that, during one hour of incubation at 90 °C, ceramide was converted into sphingosine. Samples were then separated with chloroform and water, subsequently, the upper phase was removed and the organic phase was dried under nitrogen. The level of sphingosine sourced from ceramide was determined by the HPLC method described above.

### 2.5. Analysis of Proteins Expression

The expression of proteins from the insulin signaling pathway as well as proteins involved in the sphingolipid signaling pathway was performed using the Western Blot technique, as previously described by Konstantynowicz-Nowicka et al. [21]. Firstly, the left ventricle samples were homogenized engaging radioimmunoprecipitation assay (RIPA) buffer including phosphatase and protease inhibitors (Roche Diagnostics GmbH, Manheim, Germany). The bicinchoninic acid (BCA) assay method with bovine serum albumin (BSA) used as a standard was employed to establish the total protein concentration in the prepared homogenates. Then, homogenates were diluted in Laemmli buffer (Bio-Rad, Hercules, CA, USA) and loaded on CriterionTM TGX Stain-Free Precast Gels (Bio-Rad, Warsaw, Poland) at the same protein volume. After that, the proteins during electrophoresis were separated and directly transferred onto nitrocellulose membranes. In the next step, the membranes were blocked in the tris-buffered saline buffer with Tween-20 (TBST) containing 5% non-fat dry milk or 5% BSA and, next, immunoblotted overnight at 4 °C with appropriate primary antibodies: glycogen synthase kinase 3β (GSK3β, 1:500, Thermo Scientific, Rockford, IL, USA), phosphorylated glycogen synthase kinase 3β (Ser9) (pGSK3β (Ser9), 1:500; Thermo Scientific, USA), insulin receptor substrate 1 (IRS1, 1:1000; Cell Signaling Technology, Danvers, MA, USA), phosphorylated insulin receptor substrate 1 (Ser302) (pIRS1 (Ser302), 1:1000; Cell Signaling Technology), AS160 protein (AS160, 1:500; Cell Signaling Technology), phosphorylated AS160 protein (Thr642) (pAS160 (Thr642), 1:500; Cell Signaling Technology), protein kinase B (Akt/PKB, 1:500; Cell Signaling Technology), phosphorylated protein kinase B (Ser473) (pAkt (Ser473), 1:500; Cell Signaling Technology), glucose transporter 1 (GLUT1, 1:500; Santa Cruz Biotechnology, Inc., Dallas, TX, USA), glucose transporter 4 (GLUT4, 1:500; Santa Cruz Biotechnology), serine palmitoyltransferase subunit 1 (SPTLC1, 1:5000, Abcam, Cambridge, UK), serine palmitoyltransferase subunit 2 (SPTLC2, 1:500, Santa Cruz Biotechnology), alkaline sphingomyelinase (Alk-SMase, 1:500, Santa Cruz Biotechnology), neutral sphingomyelinase (N-SMase, 1:500, Santa Cruz Biotechnology), sphingosine kinase 1 (SPHK1, 1:500, Sigma-Aldrich), sphingosine kinase 2 (SPHK2, 1:500, Sigma-Aldrich, Saint Louis, MO, USA), acid ceramidase (ASAH1, 1:500, Santa Cruz Biotechnology), neutral ceramidase (ASAH2, 1:500, Santa Cruz Biotechnology), alkaline ceramidase (ASAH3, 1:500, Thermo Scientific), ceramide synthase 4 (LASS4, 1:500, Sigma-Aldrich), ceramide synthase 5 (LASS5, 1:500, Thermo Scientific), ceramide synthase 6 (LASS6, 1:500, Thermo Scientific). Next, nitrocellulose membranes were incubated with horseradish peroxidase (HRP, Bio-Rad) conjugated with proper secondary antibodies. Subsequently, the chemiluminescence substrate (Clarity Western ECL Substrate, Bio-Rad, Hercules, CA, USA) was used in order to visualize protein bands and the obtained signals were quantified densitometrically by a ChemiDoc visualization system (Image Laboratory Software, Bio-Rad, Warsaw, Poland). The expression of proteins was standardized to the total protein expression and the control was set at 100%.

### 2.6. Statistical Analysis

All data are expressed as the mean ± standard deviation (SD) and based on ten (sphingolipid analysis) or six (Western Blot) independent determinations in each experimental group. The distribution of values and homogeneity of the variance were assessed employing the Shapiro–Wilk test and Bartlett’s test. Statistical comparisons were performed by one-way ANOVA followed by a respective post hoc test (Tukey’s test and *t*-test) using GraphPad Prism 5 (GraphPad Software, San Diego, CA, USA). The statistical significance was defined as *p* < 0.05.

## 3. Results

### 3.1. Effects of Eight-Week NAC Treatment on Body Weight, Plasma Glucose and Insulin Concentrations as Well as HOMA-IR in Rats Subjected to Standard and High-Fat Diets

In our basic metabolic parameter, body weight was significantly increased in lipid overload conditions (+23.9%, *p* < 0.05, vs. control group; Appendix A). We showed a significant change in the body weight in the NAC group (−22.8%, *p* < 0.05, vs. HFD group; Appendix A) as well as in rats subjected to HFD with NAC treatment (+13.3%, *p* < 0.05, vs. control group; −8.5%, *p* < 0.05, vs. HFD group; Appendix A). We established that the glucose level in the HFD group was higher (+10.0%, *p* < 0.05; Appendix A) than the control group. There was a relevant impairment in the glucose level in both NAC-treated groups (NAC: −15.4%, *p* < 0.05, vs. HFD group; HFD+NAC: −14.2%, *p* < 0.05, vs. HFD group; Appendix A). Moreover, we reveled a pronounced increase in plasma insulin level in the HFD group (+98.4%, *p* < 0.05, vs. control group; Appendix A). Additionally, there were significant changes in the NAC group (+35.8%, *p* < 0.05, vs. control group; −31.6%, *p* < 0.05, vs. HFD group; Appendix A) as well as the HFD+NAC group (+73.3%, *p* < 0.05, vs. control group; −12.7%, *p* < 0.05, vs. HFD group; Appendix A). As expected, the high-fat diet caused an increase in the HOMA-IR value (+79.4%, *p* < 0.05, vs. control group; Appendix A). Moreover, we noticed a significant alternation in the value of HOMA-IR in the NAC group (−43.3%, *p* < 0.05, vs. HFD group; Appendix A) as well as the HFD+NAC group (+61.6%, *p* < 0.05, vs. control group; −9.9%, *p* < 0.05, vs. HFD group; Appendix A).

### 3.2. Effects of Eight-Week ALA Treatment on Body Weight, Plasma Glucose and Insulin Concentrations as Well as HOMA-IR in Rats Subjected to Standard and High-Fat Diets

Our data demonstrated a significant increase in the final body weight in the HFD group (+33.9%, *p* < 0.05, vs. control group; Appendix A). We showed significant changes in the body weight in the ALA group (−26.6%, *p* < 0.05, vs. HFD group; Appendix A) as well as the HFD+ALA (+18.7%, *p* < 0.05, vs. control group; −11.4%, *p* < 0.05, vs. HFD group; Appendix A). The glucose plasma level was higher in the HFD group (+9.1%, *p* < 0.05, vs. control group; Appendix A). There were alterations in glucose concentrations in both ALA-treated groups (ALA: −13.2%, *p* < 0.05, vs. HFD group; HFD+ALA: −10.0%, *p* < 0.05, vs. HFD group; Appendix A). Moreover, in the HFD group, we noticed an increased concentration of insulin (+85.6%, *p* < 0.05, vs. control group; Appendix A). There were significant changes in the ALA group (−47.4%, *p* < 0.05, vs. HFD group; Appendix A) as well as the HFD+ALA group (+52.7%, *p* < 0.05, vs. control group; −17.8%, *p* < 0.05, vs. HFD group; Appendix A). Interestingly, we observed a considerable elevation the HOMA-IR value after the HFD course (+87.9%, *p* < 0.05, vs. control group; Appendix A). In addition, we revealed significant changes in the HOMA-IR value in the ALA group (−47.7%, *p* < 0.05, vs. HFD group; Appendix A) as well as the HFD+ALA group (+59.0%, *p* < 0.05, vs. control group; −15.4%, *p* < 0.05, vs. HFD group; Appendix A).

### 3.3. Effects of Eight-Week NAC Treatment on Sphingolipid Concentrations in Plasma of Rats Subjected to Standard and High-Fat Diets

In our study, we observed a substantial increase in plasma levels of sphinganine (SFA) in each experimental group (HFD: +67.3%, NAC: +53.1%, HFD+NAC: +31.5%, *p* < 0.05; Figure 1A) compared to the control rats. As expected, eight-week NAC administration to rats fed the high-fat diet caused a considerable decrease in the concentration of SFA (−21.4%, *p* < 0.05; Figure 1A) in comparison with the HFD group. In the HFD group treated with NAC, a significant elevation in the sphinganine-1-phosphate (SA1P) content (+39.5%, *p* < 0.05, vs. control group; +68.8%, *p* < 0.05, vs. HFD group; Figure 1B) was observed. Moreover, there were significant differences in plasma levels of ceramide (CER) (−34.1%, *p* < 0.05, vs. control group; −37.7%, *p* < 0.05, vs. HFD group; Figure 1C) in rats treated by NAC alone. Furthermore, the sphingosine (SFO) concentration in the HFD group was higher (+9.2%, *p* < 0.05; Figure 1D) compared to the control rats. The chronic NAC administration to rats fed HFD substantially increased the content of SFO (+35.5%, *p* < 0.05, vs. control group; +24.0%, *p* < 0.05, vs. HFD group; Figure 1D). In the HFD+NAC group, the level of sphingosine-1-phosphate (S1P) was lower (+35.5%, *p* < 0.05; Figure 1E) than the control group. Our study demonstrated a rise in the value of the S1P/CER ratio in the following groups, i.e., HFD (+22.0%, *p* < 0.05, vs. control group; Figure 1F), NAC (+66.2%, *p* < 0.05, vs. control group; Figure 1F) as well as HFD+NAC (+63.0%, *p* < 0.05, vs. control group; +33.6%, *p* < 0.05, vs. HFD group; Figure 1F).

### 3.4. Effects of Eight-Week ALA Treatment on Sphingolipid Concentrations in Plasma of Rats Subjected to Standard and High-Fat Diets

Overall, we noticed an increase in plasma SFA content in each experimental group (HFD: +52.0%, ALA: +190.7%, HFD+ALA: +131.5% *p* < 0.05; Figure 1A) in comparison with the control group. Additionally, the SFA concentration in both ALA groups was higher (ALA: +91.2%, HFD+ALA: +52.2%, *p* < 0.05; Figure 1A) than the appropriate HFD group. Furthermore, the chronic ALA application to rats fed the standard diet considerably increased the concentrations of SA1P (+47.6%, *p* < 0.05, vs. control group; +58.9%, *p* < 0.05, vs. HFD group; Figure 1B). Concomitantly, the HFD-induced obesity group treated with ALA disclosed a relevant increment content of SA1P (+76.1%, *p* < 0.05, vs. control group; +89.5%, *p* < 0.05, vs. HFD group; Figure 1B). In the ALA alone group, significant elevation in the CER content (+34.1%, *p* < 0.05, vs. control group; +44.1%, *p* < 0.05, vs. HFD group; Figure 1C) was demonstrated. In the HFD group, the level of SFO was lower (+50.3%, *p* < 0.05; Figure 1D) compared to the control group. However, the SFO concentration in the ALA alone group was lower (−31.0%, *p* < 0.05; Figure 1D) than the proper HFD group. Our study demonstrated that the HFD-induced obesity group treated with ALA exhibited a significant elevation in SFO concentrations (+96.8%, *p* < 0.05, vs. control group; +31.0%, *p* < 0.05, vs. HFD group; Figure 1D). Moreover, in both ALA-treated groups, we observed a substantial rise in the content of S1P (ALA: +46.0%, *p* < 0.05, vs. control group; +30.4%, *p* < 0.05, vs. HFD group; HFD+ALA: +56.8%, *p* < 0.05, vs. control group; +40.0%, *p* < 0.05, vs. HFD group; Figure 1E). Additionally, compared to the standard conditions, the value of the S1P/CER ratio was significantly increased in the following experimental groups, i.e., HFD and HFD+ALA (+22.6%, +55.8%, *p* < 0.05, respectively; Figure 1F). In the ALA alone group, the level of the S1P/CER ratio was markedly reduced (−11.1%, *p* < 0.05; Figure 1F) in comparison with the HFD group.

### 3.5. Effects of Eight-Week NAC Treatment on Sphingolipid Concentrations in the Left Ventricle of Rats Subjected to Standard and High-Fat Diets

In our study, the high-fat feeding resulted in a significant increment in the myocardial concentration of SFA (+54.5%, *p* < 0.05; Figure 2A) compared to the control group. However, the rats fed the standard chow with NAC application exhibited substantially reduced concentrations of SFA (−30.7%, *p* < 0.05; Figure 2A) in relation to the appropriate HFD group. In the HFD group, the level of CER was higher (+19.1%, *p* < 0.05; Figure 1C) than the proper control group. Concomitantly, in both NAC treated groups was observed an appreciable reduction in the content of CER (NAC: −14.4%, *p* < 0.05; HFD+NAC: −13.7%, *p* < 0.05; Figure 1C) in comparison to the HFD group. In the HFD group, we observed a significant elevation in the SFO content (+31.9%, *p* < 0.05; Figure 1B) compared to the control group. However, the rats fed the standard chow with NAC administration disclosed substantially reduced SFO concentrations (−19.1%, *p* < 0.05, vs. control group; −38.7%, *p* < 0.05, vs. HFD group; Figure 2D). We also noticed a marked reduction in the level of S1P (−42.3%, *p* < 0.05; 229 Figure 2E) under the lipid overload conditions. Furthermore, in both NAC-treated groups, we demonstrated an increase in the level of S1P (NAC: +78.1%, *p* < 0.05; HFD+NAC: +64.0%, *p* < 0.05; Figure 2E) in relation to the appropriate HFD group. In the HFD group, we also observed a significant diminishment of the S1P/CER ratio value (−43.8%, *p* < 0.05; Figure 2F) compared to the control rats. In both NAC-treated groups, we also noticed an increase in the level of the S1P/CER ratio (NAC: +66.0%, *p* < 0.05; HFD+NAC: +70.1%, *p* < 0.05; Figure 2F) compared to the corresponding HFD group.

### 3.6. Effects of Eight-Week ALA Treatment on Sphingolipid Concentrations in the Left Ventricle of Rats Subjected to Standard and High-Fat Diets

In the left ventricle, we observed a significant rise in the SFA content in the following groups, i.e., HFD (+74.6%, *p* < 0.05; Figure 2A) and ALA (+19.0%, *p* < 0.05; Figure 2A) in relation to the appropriate control group. However, ALA treatment both in the rats fed a standard and fat chow-rich diet resulted in a significant decrease in SFA (−31.8%, −40.3%, *p* < 0.05, respectively; Figure 2A) compared to the appropriate HFD group. Concomitantly, we noticed a marked reduction in the level of SA1P in two groups, i.e., HFD (−29.5%, *p* < 0.05, vs. control group; Figure 2B) as well as ALA (−31.3%, *p* < 0.05, vs. control group; Figure 2B). In the HFD+ALA group, the level of SA1P was higher (+69.4%, *p* < 0.05; Figure 2B) than the proper HFD group. Furthermore, in the HFD group, the level of CER was greater (+11.9%, *p* < 0.05; Figure 1E) than the proper control group. Moreover, the HFD group with ALA administration was characterized by a substantial reduction in the concentration of CER (−6.7%, *p* < 0.05, vs. control group; −16.6%, *p* < 0.05, vs. HFD group; Figure 2C). In the HFD group, the level of SFO was lower (+46.5%, *p* < 0.05; Figure 1D) compared to the control group. In ALA alone and HFD+ALA groups, we noticed a significant decrease in the content of SFO (−34.2%, −35.4%, *p* < 0.05, respectively; Figure 2D) compared to the HFD group. In our study, we observed a substantial decrease in the left ventricle levels of S1P in each experimental group (HFD: −51.5%, ALA: −19.9%, HFD+ALA: −19.9%, *p* < 0.05; Figure 2E) compared to the control rats. However, in both ALA-treated groups, we observed an elevated content of S1P (ALA: +65.2%, *p* < 0.05; HFD+ALA: +65.2%, *p* < 0.05; Figure 2E) in comparison to the proper HFD group. Our data demonstrated a decrease in the value of the S1P/CER ratio in the following groups, i.e., HFD (−50.6%, *p* < 0.05; Figure 2F) and ALA (−22.4%, *p* < 0.05; Figure 2F) compared to the control group. In both ALA-treated groups, we also demonstrated an increased value of the S1P/CER ratio (ALA: +57.2%, *p* < 0.05; HFD+ALA: +81.9%, *p* < 0.05; Figure 2F) in relation to the HFD group.

### 3.7. Effects of Eight-Week NAC and ALA Treatment on the Total Expression of Proteins Involved in the Sphingolipid Pathway in the Left Ventricle of Rats Subjected to Standard and High-Fat Diets

#### 3.7.1. NAC

We measured the total expression of proteins from the ceramide de novo synthesis pathway, i.e., serine palmitoyltransferase subunits 1 and 2 (SPTLC1 and SPTLC2) and ceramide synthase isoforms 4, 5, and 6 (LASS4, LASS5, and LASS6). In our study, we showed an increase the expression of SPTLC1 in each experimental group (HFD: +29.7%, NAC: +28.7%, HFD+NAC: +70.8%, *p* < 0.05; Figure 3A) compared to the control rats. In addition, the expression of SPTLC1 in the HFD+NAC group was higher (+31.7%, *p* < 0.05; Figure 3A) than in the HFD group. We observed that the high-fat diet subjected caused a significant increase in the total expression of SPTLC2 (+38.6%, *p* < 0.05, vs. control group; Figure 3B). However, in the HFD+NAC group, the expression of SPTLC2 was lower (−23.3%, *p* < 0.05; Figure 3B) than in the HFD group. In the HFD group, we observed a marked elevation in the expression of LASS5 (+26.8%, *p* < 0.05, vs. control group; Figure 3D). Additionally, the total expression of LASS5 was markedly reduced in the NAC group (NAC: −39.0%, *p* < 0.05, vs. control group; −51.9%, *p* < 0.05, vs. HFD group; Figure 3D) as well as the HFD+NAC group (HFD+NAC: −29.3%, *p* < 0.05, vs. control group; −44.2%, *p* < 0.05, vs. HFD group; Figure 3D). Moreover, we did not notice any significant difference in the total expression of LASS4 and LASS6 (*p* > 0.05; Figure 3C,E).

#### 3.7.2. ALA

Obesity induced by HFD resulted in increased expression of SPTLC1 (+34.4%, *p* < 0.05; Figure 3A) compared to the control rats. On the other hand, we demonstrated that chronic ALA treatment in rats fed the standard chow and the high-fat diet considerably reduced the expression of SPTLC1 (−33.9%, −31.3%, *p* < 0.05, respectively; Figure 3A) compared to the HFD group. In the case of SPTLC2 expression, we obtained similar results in particular groups: HFD (+29.2%, *p* < 0.05, vs. control group; Figure 3B), ALA (−18.0%, *p* < 0.05, vs. HFD group; Figure 3B), HFD+ALA (−15.3%, *p* < 0.05, vs. HFD group). In both ALA-treated groups, LASS4 expression was significantly decreased (ALA: −36.1%, *p* < 0.05; HFD+ALA: −22.4%, *p* < 0.05; Figure 3C) in comparison to the HFD group. In the case of LASS5 expression, a substantial increment was detected in rats subjected to the HFD (+35.2%, *p* < 0.05, vs. control group; Figure 3C). In the case of LASS6 expression, the alternations did not reach a significant level (*p* > 0.05; Figure 3E).

#### 3.7.3. NAC

Overall, we analyzed the total expression of proteins from the salvage pathway, i.e., sphingosine kinase 1 and 2 (SPHK1 and SPHK2), acid ceramidase (ASAH1), neutral ceramidase (ASAH2), and alkaline ceramidase (ASAH3). We observed that the total expression of ASAH1 in the HFD group was higher (+31.4%, *p* < 0.05; Figure 4C) than the rats fed a standard diet. The total expression of ASAH2 was considerably reduced in the NAC group (−45.1%, *p* < 0.05, vs. control group; −51.9%, *p* < 0.05, vs. HFD group; Figure 4D) as well as in the HFD+NAC group (−30.5%, *p* < 0.05, vs. control group; −39.1%, *p* < 0.05, vs. HFD group; Figure 4D). We did not observe any significant differences in the total expression of SPHK1, SPHK2, and ASAH3 (*p* > 0.05; Figure 4A,B,E).

#### 3.7.4. ALA

We demonstrated significant changes in the total expression of ASAH1 in rats fed the high-fat diet (+59.1%, *p* < 0.05, vs. control group; Figure 4C) as well as in rats subjected to HFD with NAC (+21.4%, *p* < 0.05, vs. control group; −23.7%, *p* < 0.05, vs. HFD group; Figure 4C). We did not notice any significant differences in the total expression of SPHK1, SPHK2, ASAH2, and ASAH3 (*p* > 0.05; Figure 4A,B,D,E).

#### 3.7.5. NAC

We also determined the total expression of proteins involved in the sphingomyelinase pathway, i.e., alkaline and neutral sphingomyelinase (Alk-SMase and N-SMase). In both NAC treated groups, we observed a significant decrease in the expression of Alk-SMase (NAC: −41.4%, *p* < 0.05, vs. HFD group; HFD+NAC: −29.8%, *p* < 0.05, vs. HFD group; Figure 5A). Moreover, we showed an increase in the expression of N-SMase in the HFD group (+26.1%, *p* < 0.05; Figure 5B) compared to the control rats. However, the expression of N-SMase in the HFD+NAC was lower (−26.6%, *p* < 0.05; Figure 5B) than in the HFD group.

#### 3.7.6. ALA

We observed that the high-fat diet caused a significant increase in the total expression of N-SMase (+31.3%, *p* < 0.05; Figure 5B) in relation to the control rats. Furthermore, in both ALA-treated groups, the expression of N-SMase was lower (ALA: −18.9%, *p* < 0.05; −25.1%, *p* < 0.05; Figure 5B) than in the HFD group. In the case of Alk-SMase expression, the alternations did not reach a significant level (*p* > 0.05; Figure 5A).

### 3.8. Effects of Eight-Week NAC and ALA Treatment on the Total Expression of Glucose Transporters and Phosphorylation State of Proteins Involved in Insulin Signaling Pathway in the Left Ventricle of Rats Subjected to Standard and High-Fat Diets

#### 3.8.1. NAC

As expected, the phosphorylation of protein kinase B (Akt) (Ser473) decreased after the HFD course (−37.1%, *p* < 0.05, vs. control group; Figure 6B). What is more, in both NAC treated groups (the standard chow and the high-fat diet), we observed a pronounced increase in the phosphorylation of Akt (Ser473) in the NAC group (+88.9%, *p* < 0.05, vs. HFD group; Figure 6B) and the HFD+NAC group (+36.1%, *p* < 0.05, vs. control group; +116.4%, *p* < 0.05, vs. HFD group; Figure 6B). Concomitantly, in the HFD group, we found a significant decrease in the phosphorylation of glycogen synthase 3β (GSK3β) (Ser9) (−51.4%, *p* < 0.05, vs. control group; Figure 6D). In addition, a significant increase in the phosphorylation of GSK3β (Ser9) was observed in the NAC group (+98.6%, *p* < 0.05; Figure 6D) as well as in the HFD+NAC group (+145.4%, *p* < 0.05; Figure 6D) in relation to the HFD group. Rats subjected to HFD showed a substantial decrease in the total expression of glucose transporter 1 and 4 (GLUT1 and GLUT4) (−34.6%, −55.2%, *p* < 0.05, respectively; Figure 6E,F) compared to the corresponding control groups. Additionally, eight-week NAC treatment resulted in substantial differences in the expression of GLUT4 in rats fed standard chow (NAC: −36.1%, *p* < 0.05, vs. control group; +42.4%, *p* < 0.05, vs. HFD group; Figure 6F) and rats fed high-fat chow (HFD+NAC: −41.7%, *p* < 0.05, vs. control group; +30.1%, *p* < 0.05, vs. HFD group; Figure 6F). However, the phosphorylation of insulin receptor substrate 1 (IRS1) (Ser302) and AS160 protein (AS160) (Thr642) remained unchanged in each experimental group (*p* > 0.05; Figure 6A,C).

#### 3.8.2. ALA

In the experimental model of HFD-induced obesity, we observed a considerable reduction in the phosphorylation of Akt (Ser473) (−27.1%, *p* < 0.05, vs. control group; Figure 6B). Moreover, in both ALA-treated groups, we showed significant elevation in the phosphorylation of Akt (Ser473) (ALA: +56.5%, *p* < 0.05, vs. HFD group; HFD+ALA: +47.5%, *p* < 0.05, vs. HFD group; Figure 6B). In the HFD group, we noticed a decrease in the phosphorylation of GSK3β (Ser9) (−42.2%, *p* < 0.05, vs. control group; Figure 6D). There were significant differences in GSK3β phosphorylation (Ser9) in both ALA-treated groups (ALA: −33.3%, *p* < 0.05, vs. control group; HFD+ALA: +72.8%, *p* < 0.05, vs. HFD group; Figure 6D). In the HFD-induced obesity group, we noticed a marked reduction in the total expression of GLUT1 (−56.7%, *p* < 0.05, vs. control group; Figure 6E) as well as a significant decline in the HFD+ALA group (−49.1%, *p* < 0.05, vs. control group; Figure 6E). Concomitantly, in the HFD group, we presented a decrease in the total expression of GLUT4 (−15.7%, *p* < 0.05, vs. control group; Figure 6F). Interestingly, chronic ALA administration with high-fat feeding caused a significant elevation in the total expression of GLUT4 (+32.6%, *p* < 0.05, vs. HFD group; Figure 6F). Similar to NAC treatment, we did not found any significant alternations in the phosphorylation of IRS1 (Ser302) and AS160 (Thr642) (*p* > 0.05; Figure 6A,C).

## 4. Discussion

Numerous studies suggest that excessive fatty acids uptake is a crucial factor in the alteration of lipid metabolism and it contributes to the development of metabolic cardiac diseases. Cardiac muscle cells have a restricted capacity for lipid storage, allowing for the proper functioning of the tissue. In enhanced FA conditions, this ability is exceeded, favoring ectopic lipid accumulation and the occurrence of cardiac lipotoxicity, which contributes to the alteration of its structure and function. Thus, the optimization of energy harvested from lipid metabolism may be important in the improvement of heart dysfunction [22]. Hence, in our study, we elucidated the potential cardioprotective role of two selected antioxidants, i.e., α-lipoic acid and n-acetylcysteine, on altered myocardial sphingolipid levels coexisting with insulin resistance induced by a diet rich in fat.

Increased circulating fatty acids, resulting from HFD, are initially stored in TAG but the surplus of FAs are shunted to pathways leading to the synthesis of non-oxidative molecular mediators, such as ceramide, from which it may the ensuing lipotoxic effects [23,24]. It is known that the toxic effects of FAs are mainly due to their saturation and chain length. Hence, saturated long-chain fatty acids (LCFAs) such as palmitic acid and stearic acid appear to be the most toxic [25]. In a model of rats, ceramide accumulation was a consequence of increased uptake and esterification of LCFAs into myocytes [26]. This is consistent with our observations, which demonstrated that rats fed an HFD for 8 weeks exhibited enhanced accumulation of ceramide in the heart resulting in exacerbation of all three pathways of ceramide synthesis, i.e., de novo, salvage, and sphingomyelinase pathways. However, in our opinion, de novo is the most dominant due to a significant surplus of cytoplasmic FAs in the diet. Paumen et al. reported that the murine hematopoietic cells exposed to LCFAs increased ceramide de novo synthesis, which was exacerbated by the inhibition of β-oxidation of fatty acids via break carnitine palmitoyltransferase I (CPT I) expression leading to further palmitate-induced CER synthesis and cell death [27]. Even more interesting, our study revealed that NAC and ALA administration inhibited myocardial ceramide production and reduced the obesity effects at different stages of the sphingolipid pathway. We observed that NAC treatment caused a decrement in the expression of proteins regulating all three ceramide synthesis pathways, i.e., SPTLC2, LASS5, ASAH2, Alk-SMase, and N-SMase, which indicates a significant inhibiting influence on the cardiac ceramide synthesis normalizing the HFD-induced effects. Our observations are consistent with the results of research conducted on the mouse model, where NAC induced impairment in hepatic expression of genes regulating de novo the CER pathway (SPTLC subunits 1 and 2, and ceramide synthase isoforms 4, 5, and 6) in enhanced lipid availability conditions [28]. We suppose that the protective impact of NAC on reducing the lipids level is associated with its antioxidant properties and increment of antioxidant glutathione (GSH) concentration. Furthermore, n-acetylcysteine supplementation provoked an increase in hepatic fatty acids uptake from circulation, which decreased FAs availability for peripheral tissues such as the heart muscles [29]. Elevated GSH synthesis, after NAC administration, mediates exacerbated mitochondrial β-oxidation of FAs. Moreover, NAC showed appreciable influence on HFD-related development of inflammation by inhibited expression of nuclear factor-kappa beta (NF-κβ) and changed in various cytokine syntheses [30]. In the heart tissue, NAC caused normalization of lipid changes induced by a high-sucrose diet in rats. There has been an increased degradation of myocardial FAs caused by increased activity of β-hydroxylacyl coenzyme-A dehydrogenase (OHADH), which resulted from NAC administration. In this model, NAC reduced energy expenditure from carbohydrates in favor of lipids oxidation, thus decreasing the anaerobic metabolism in the cardiac tissue [22]. In our study, simultaneously with NAC treatment, we observed that ALA, which is also a GSH precursor, decreased selected proteins expression regulating ceramide synthesis in de novo, salvage, and sphingomyelinase pathways, which further led to reversed HFD-induced ceramide accumulation. This is in agreement with a study conducted on young and old rats showing that ALA mediated increased GSH levels inducing a decline in N-SMase activity in the aortic endothelium [31]. Hence, elevated GSH levels and lower N-SMase activity have been involved in the loss of endothelial ceramide content in old animals by ALA administration [31,32,33]. Apart from the above direct ALA influence on ceramide synthesis by regulating N-SMase activity, the literature data also reported an indirect action of this acid, which improved mitochondrial efficiency and its metabolic function in lipid oxidation [31,34]. This notion suggests that ALA redirected excessive FAs supply with the diet to β-oxidation, which lowered the FFAs level and impaired de novo ceramide synthesis, regardless of the modulation of protein expression from the sphingolipid pathway [31]. Moreover, Kim et al. reported that ALA administration led to FFAs redistribution into the cell membrane and alteration of free cholesterol distribution, which may need to maintain influence on cell damage due to enhanced FAs storing [34]. Under the aforementioned data, we confirm that ALA treatment can prevent cardiac dysfunction resulting from HFD-induced lipotoxicity development.

Several studies conducted on animal models demonstrated that metabolites of fatty acids interfere in decreasing insulin transduction at multiple levels, especially the decline in phosphorylation of Akt as a consequence of the development of obesity in heart tissue in rodents [35,36,37]. In our study, we evaluated the Akt pathway as a pivotal marker of insulin signaling, which is regulated by not only insulin but also nutrition and redox status. Aburasayn et al. and Asrich et al. reported that saturated fatty acids, especially palmitic acid, indirectly through increasing CER production, negatively affected insulin action and impaired insulin signaling transduction by inhibiting the phosphorylation of Akt (Ser473 and Thr308) necessary for the activation of the Akt pathway. These, in turn, lead to decreased regulation of GLUT4 transduction into the membrane and then glucose uptake into the cells [6,38]. The above-described evidence is consistent with our observation that chronic exposure to an HFD caused a reduction in the activity of Akt (as evidenced by pAkt(Ser473)/Akt ratio) and the expression of insulin-dependent GLUT4. We suspect that the observed decline in GLUT1 with GLUT4 expression may have altered the overall transport and availability of glucose for cells in the heart tissue. Interestingly, in our study, intragastrical supplementation of antioxidants attenuated the alterations caused by IR development (following high-fat feeding of rats). Rodrigues et al., in their study on obese mice, showed that NAC raised the protein level of phosphorylated Akt in the cardiac tissue, and thus the pAkt level was restored to values close to that of the control mice [36]. A study conducted by Johnson et al. revealed that cardiomyocytes exposed to a high-glucose concentration with n-acetylcysteine showed improved glucose utilization caused by enhanced GLUT4 redistribution to the membrane allowing regulation and maintained proper glucose homeostasis [37]. In our study, we demonstrated that the diet-induced decrease in the heart pAkt(Ser473)/Akt ratio was completely reestablished. What is more, NAC, by regulating the Akt pathway, mediated the observed rise in expression of GLUT4 in the HFD+NAC group. The precise mechanism of the NAC action is still unclear, but it is known that the development of inflammation during obesity induced by an HFD is associated with increased oxidative stress, which also worsens insulin signaling and accelerates myocardial damage. Thus, in our research, NAC probably resulted in reduced ROS generation that interfere with cellular processes, i.e., the degradation of membrane lipids (lipid peroxidation) and thereby increasing the metabolic activity of the tissue and reducing cellular apoptosis [36,37]. Similarly, we also observed that ALA reversed HFD-induced insulin resistance. Our results suggest that the improvement in the insulin pathway, mainly a direct activation of insulin pathway intermediates such as Akt and GSK3β, might have a significant impact on increased basal glucose transport by enhanced expression of GLUT4 and reduced synthesis and deposition of glycogen in muscles. These observations are consistent with research conducted by Gupte et al., where ALA improved glucose uptake in the soleus muscle through partially restored phosphorylation of Akt in high-fat-fed rats [39]. Recent in vitro studies have suggested that in response to ALA administration, the activity of proteins, i.e., IRS1, PI3K, and Akt, was increased. The activation of proteins from the insulin signaling pathway also resulted in enhanced translocation of GLUT1 and 4 into membranes in adipocytes and skeletal muscle cells [40,41,42]. Furthermore, Tardif et al. and Smith et al. reported that ALA may also act indirectly on insulin-sensitive mechanisms by decreasing endothelial ceramide levels, which reduces PPA2 activity and subsequent Akt dephosphorylation [31,33]. It is also possible that ALA, by activating AMP-activated protein kinase, enhanced FAs oxidation and glucose transport, which additionally reduced cardiac IR [39].

To summarize, in metabolic pathologies evoked by an HFD, the optimization of cardiac lipid metabolism may be a crucial target for prevention and treatment of heart dysfunction limiting the cardiac ceramidose. Current studies clearly demonstrate that the antioxidants, i.e., NAC and ALA, had a beneficial effect on metabolism in obesity-related disorders, which favorably affects insulin action and improves IR. We indicated that antioxidants reduced ceramide production by alteration in the expression of enzymes from the de novo, salvage, and sphingomyelinase pathways, which may be essential for preventing metabolic pathologies and reducing cardiac insulin resistance. However, further studies will be needed in order to investigate a precise mechanism of the above-mentioned agents on lipid metabolism under insulin resistance conditions and studies should discuss the results and how they can be interpreted from the perspective of previous studies and of working hypotheses. The findings and their implications should be discussed in the broadest context possible. Future research directions may also be highlighted.

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
