# Peer review of "Antioxidants Supplementation Reduces Ceramide Synthesis Improving the Cardiac Insulin Transduction Pathway in a Rodent Model of Obesity"

_nutrients, 2021, doi:10.3390/nu13103413_

Round 1

Reviewer 1 Report

Katarzyna Hodun et al. present a well designed study and layout on a high yield topic. Manuscript is well written and contain the informative finding. 

Author Response

Białystok, 2021-09-24

Dear Madam, Dear Sir,

We appreciate the time and effort that you dedicated to providing your valuable feedback on our manuscript entitled "Antioxidants Supplementation Reduces Ceramide Synthesis Improving the Cardiac Insulin Transduction Pathway in a Rodent Model of Obesity." (authors: Katarzyna Hodun, Klaudia Sztolsztener, Adrian Chabowski). We are grateful for your insightful comments on our paper. Furthermore, the whole manuscript was corrected and improved grammatically. The changes within the manuscript have been highlighted on red color.

Here are our response to the Reviewer’s comments:

Katarzyna Hodun et al. present a well designed study and layout on a high yield topic. Manuscript is well written and contain the informative finding. 

Thank You for this comment. We are pleased with this opinion about our research.

Taking into account all the Reviewer’ suggestions and concerns provided, we believe that the manuscript was further improved and is now more suitable for publication in the Nutrients.

Yours faithfully,

Katarzyna Hodun

Department of Physiology,

Medical University of Białystok,

15-222 Białystok, Mickiewicz Str. 2 C, Poland

Email: katarzyna.hodun@umb.edu.pl

Telephone: +48857485585

FAX: + 48857485586

Reviewer 2 Report

The paper by Katarzyna Hodun et al., presents the beneficial effect of an antioxidant supplementation strategy for reducing ceramide synthesis and improving the cardiac insulin transduction pathway.

The introduction is short, and it is suffering of a lack of details, especially about the bioactive lipids pool which will be analyzed and discussed in the paper. Indeed, the word "sphingolipids" is briefly cited twice in the introduction and all the different sphingoid bases are listed quickly in the material and methods without describing their role or effects.

The study is interesting but, unfortunately, the result part is really disorganized and difficult to read. Moreover, the protein involved in the sphingolipid pathways are just appearing in the result part 3.3 to result part 5, without their complete name and their role will be described only in the discussion.

The discussion is nicely written and increases the overall scientific soundness of this paper.

There are several issues, mentioned above and below, that definitely need to be addressed and modified by the authors for publication.

Major Comments

  1. The introduction must be more detailed with explanations about the different sphingoid bases the authors decided to analyze.
  2. Results part: Overall, the authors shouldn’t merge the sphingoid bases concentrations from the NAC experiments with the ALA experiments.
  3. Results part: Overall, the figure order should follow the text (ex figure 1, the authors describe first fig1A, 1D, 1F, then 1B, then come back to 1A, 1C….)
  4. Results part: line 209, the authors can not write “in rats receiving the ALA alone we observe a significant decrease in SFO concentration (…) in comparison with the proper HFD group”, it is misleading the reader, they should write the SFO concentration in the ALA alone group is lower than the proper HFD group.
  5. Results part: the western blot pictures fig3A, fig4A, fig4C, fig5A and fig5B, can not be published because we can’t see anything. This comment is especially applying for the fig4C, fig5A and fig5B, since the authors claim statistical differences between the conditions

Minor Comments

  1. Material and methods, line 85 : the authors didn’t describe the “intragastric      administration of antioxidants”. Do they perform it via a gavage, a catheter, or a probe?

Author Response

Białystok, 2021-09-24

Dear Madam, Dear Sir,

We appreciate the time and effort that you dedicated to providing your valuable feedback on our manuscript entitled "Antioxidants Supplementation Reduces Ceramide Synthesis Improving the Cardiac Insulin Transduction Pathway in a Rodent Model of Obesity." (authors: Katarzyna Hodun, Klaudia Sztolsztener, Adrian Chabowski). We are grateful for your insightful comments on our paper. Furthermore, the whole manuscript was corrected and improved grammatically. The changes within the manuscript have been highlighted on red color.

Here are our response to the Reviewer’s comments:

The paper by Katarzyna Hodun et al., presents the beneficial effect of an antioxidant supplementation strategy for reducing ceramide synthesis and improving the cardiac insulin transduction pathway.

The introduction is short, and it is suffering of a lack of details, especially about the bioactive lipids pool which will be analyzed and discussed in the paper. Indeed, the word "sphingolipids" is briefly cited twice in the introduction and all the different sphingoid bases are listed quickly in the material and methods without describing their role or effects.

In the Introduction section we changes some information (line number: 42-45, 62-68). We did not describe role of all sphingolipid fractions, we only mentioned selected lipids pools in order not to distract readers’ attention.

The study is interesting but, unfortunately, the result part is really disorganized and difficult to read. Moreover, the protein involved in the sphingolipid pathways are just appearing in the result part 3.3 to result part 5, without their complete name and their role will be described only in the discussion.

We agree that the result part is difficult for the reader. So, we reorganized this part and wrote results section again and added fully name of proteins from sphingolipid and insulin pathways, which will make it easier for reading our manuscript.

The discussion is nicely written and increases the overall scientific soundness of this paper.

There are several issues, mentioned above and below, that definitely need to be addressed and modified by the authors for publication.

Major Comments

  1. The introduction must be more detailed with explanations about the different sphingoid bases the authors decided to analyze.

As above-mentioned, we slightly reorganized the Introduction part.

  1. Results part: Overall, the authors shouldn’t merge the sphingoid bases concentrations from the NAC experiments with the ALA experiments.

We did as you suggested. We separated the results about the content of sphingoid bases in the plasma as well as tissue.

  1. Results part: Overall, the figure order should follow the text (ex figure 1, the authors describe first fig1A, 1D, 1F, then 1B, then come back to 1A, 1C….)

In order to short results section description we have placed similar changes in the same sentences (mainly due to the large number of results and its description). However, as the Reviewer’s suggested we reorganized some description of results to easier analyze.

  1. Results part: line 209, the authors can not write “in rats receiving the ALA alone we observe a significant decrease in SFO concentration (…) in comparison with the proper HFD group”, it is misleading the reader, they should write the SFO concentration in the ALA alone group is lower than the proper HFD group.

We agree with the Reviewer that we simplified mentioned sentence. So, this Results part we descripted again to easier form.

  1. Results part: the western blot pictures fig3A, fig4A, fig4C, fig5A and fig5B, can not be published because we can’t see anything. This comment is especially applying for the fig4C, fig5A and fig5B, since the authors claim statistical differences between the conditions

Indeed, mentioned representative blots are very poor. Sometimes images from Western blot seems to be a little bit faint. However, we changed representative image for better one.

Minor Comments

  1. Material and methods, line 85 : the authors didn’t describe the “intragastric administration of antioxidants”. Do they perform it via a gavage, a catheter, or a probe?

We agree that “intragastric administration of antioxidants” was not precise. We changed it in a sentence to “Once daily, each morning between 8-9 am, NAC (at a dose of 500 mg/kg body weight, Sigma Aldrich, A9165) and ALA (at a dose of 30 mg/kg body weight, Sigma Aldrich, PHR2561) were dissolved in saline solution and immediately applied intragastrically by gastric gavage to rats from appropriate groups.” Line number: 95.

Taking into account all the Reviewer’ suggestions and concerns provided, we believe that the manuscript was further improved and is now more suitable for publication in the Nutrients.

Yours faithfully,

Katarzyna Hodun

Department of Physiology,

Medical University of Białystok,

15-222 Białystok, Mickiewicz Str. 2 C, Poland

Email: katarzyna.hodun@umb.edu.pl

Telephone: +48857485585

FAX: + 48857485586

Reviewer 3 Report

Interesting work. However, it needs big improvements. Below my comments:

1) Provide an experimental plan flow. A schematic cartoon which provides the reader with a more clear description of the experiments performed, study groups, end-points, time-line, detailed methods, ... would help. 

2) Western blot is written with only one "t"; overall English language needs a profound revision. A native English language speaker would help.

3) I don't see any basic characterization of the rat model used. As this is an in vivo study, I do like to have more details on this regards. Any baseline and after treatments gravimetric data along with parameters of metabolic dysfunction (blood glucose levels, insulin levels, ...) would make this study more rigorous and complete. Have these parameters changed after administration of NAC and ALA? Also, please, add any cardiac specific parameters. It would be interesting to investigate how beneficial your treatments on cardiac function is. Ideally, a basic cardiac ultrasound analysis and any parameters of heart function and geometry should be included. 

Author Response

Białystok, 2021-09-24

Dear Madam, Dear Sir,

We appreciate the time and effort that you dedicated to providing your valuable feedback on our manuscript entitled "Antioxidants Supplementation Reduces Ceramide Synthesis Improving the Cardiac Insulin Transduction Pathway in a Rodent Model of Obesity." (authors: Katarzyna Hodun, Klaudia Sztolsztener, Adrian Chabowski). We are grateful for your insightful comments on our paper. Furthermore, the whole manuscript was corrected and improved grammatically. The changes within the manuscript have been highlighted on red color.

Here are our response to the Reviewer’s comments:

Interesting work. However, it needs big improvements. Below my comments:

1) Provide an experimental plan flow. A schematic cartoon which provides the reader with a more clear description of the experiments performed, study groups, end-points, time-line, detailed methods, ... would help.

We agree with the reviewer that a schematic cartoon may be helpful for understanding the experimental model and clearly presented experimental protocol. Thus, we added this information to the Materials and Methods section.

Line number: 109-111.

2) Western blot is written with only one "t"; overall English language needs a profound revision. A native English language speaker would help.

We appreciate spotting this language mistake (line number: 19,159,199).

We have changed this part. Additionally, in the whole our manuscript a profound English language revision have been made and all changes have been highlighted.

3) I don't see any basic characterization of the rat model used. As this is an in vivo study, I do like to have more details on this regards. Any baseline and after treatments gravimetric data along with parameters of metabolic dysfunction (blood glucose levels, insulin levels, ...) would make this study more rigorous and complete. Have these parameters changed after administration of NAC and ALA? Also, please, add any cardiac specific parameters. It would be interesting to investigate how beneficial your treatments on cardiac function is. Ideally, a basic cardiac ultrasound analysis and any parameters of heart function and geometry should be included.

Indeed, in Animals and Experimental Protocol section description of data was not precise. Thus, in Supplementary Materials we added more details on the body mass of rats and basic metabolic parameters in plasma, i.e., glucose and insulin levels with calculated HOMA-IR index. Moreover, the description of above-mentioned basic parameter was located in the first and second points (3.1. for NAC set and 3.2. for ALA set)  in the Results section in our manuscript. In addition in the Materials and Methods section we described the protocol of above-mentioned measurements (line number: 112-129).

We understated that cardiac specific parameters would be helpful for an overall assessment of holistic heart rate but in our study we focused on changes in the sphingolipid metabolism and our aim was metabolic alteration in cardiac dysfunction. We believe that hemodynamic parameters would enrich our study, however added more information about cardiac function and geometry to this manuscript can difficult to peruse and analyze contained results. Moreover, it sounds like a whole new study not just the piece of information to add into this work, that is why we decided to conduct such experiments in the next step.

Taking into account all the Reviewer’ suggestions and concerns provided, we believe that the manuscript was further improved and is now more suitable for publication in the Nutrients.

Yours faithfully,

Katarzyna Hodun

Department of Physiology,

Medical University of Białystok,

15-222 Białystok, Mickiewicz Str. 2 C, Poland

Email: katarzyna.hodun@umb.edu.pl

Telephone: +48857485585

FAX: + 48857485586

Round 2

Reviewer 3 Report

I’m satisfied with authors’ reply and revisions. The ms is suitable for publication.